# Observability Analysis and Observer Design for a Nonlinear Three-Tank System: Theory and Experiments

**DOI:** 10.3390/s20236738

**Published:** 2020-11-25

**Authors:** Santiago Rúa, Rafael E. Vásquez, Naveen Crasta, Carlos A. Zuluaga

**Affiliations:** 1School of Engineering, Universidad Pontificia Bolivariana, Medellín 050031, Colombia; santiago.rua@unad.edu.co (S.R.); carlos.zuluaga@upb.edu.co (C.A.Z.); 2Grupo de Investigación en Desarrollo Tecnológico GIDESTEC, Universidad Nacional Abierta y a Distancia, Carrera 45 # 55-19, Medellín 050012, Colombia; 3Institute for Systems and Robotics, Instituto Superior Técnico, 1049-001 Lisbon, Portugal; naveen.crasta@gmail.com

**Keywords:** observability analysis, high-gain observer, state estimation, advanced process control, soft sensor, three-tank system

## Abstract

This paper addresses the observability analysis and observer design for a nonlinear interacting three-tank system. The plant configuration is first described using the process and instrumentation diagram (P&ID) and a state–space realization is derived; some insights about the behavior of the nonlinear system, considering equilibrium points and the phase portrait are provided. Then, observability in the Hermann–Krener sense is analyzed. A high-gain observer (HGO) is then designed, using the equivalence of the original state–space realization with its observability canonical form, in order to guarantee convergence of the state estimation. The performance was validated through simulation and experiments in a multipurpose plant equipped with real sensors; the HGO response was compared to a Luenberger observer (for a linear approximation of the plant) and the Extended Kalman Filter (for which convergence is not guaranteed), considering nonlinearities, interaction, disturbances and noise. Theoretical and experimental results show that the HGO can provide robust estimation and disturbance rejection, despite the sensitivity of HGOs to noisy variables in processes such as level of liquids.

## 1. Introduction

Industrial processes show some characteristics that make difficult their regulation, for instance multivariable interactions between controlled and manipulated variables, non-measurable state variables, non-measurable disturbances, uncertain and time-varying parameters, restrictions in manipulated variables and nonlinearities, among others [1,2,3]. Modern control strategies involve the use of compensators that comprise a state feedback and an observer [4]. Observers are used to reconstruct the unmeasured state variables, since they are either unmeasurable (e.g., when they have no physical sense) or there is no technical and/or economical viability to use high-priced sensors, from the input–output behavior [5,6].

State observers and soft sensors are being used more often nowadays in control systems to achieve different objectives in several industries. For instance, Liu [7] presented a robust adaptive observer for multiple-input multiple-output nonlinear systems with unknown parameters, uncertain nonlinearities, disturbances and unmodeled dynamics; Ciccio et al. [8] proposed a new nonlinear observer-based controller for time-delay nonlinear systems; Fernandes et al. [9] proposed a motion control system for a underwater vehicle using a MIMO PID controller aided by a high-gain observer; Turki et al. [10] developed a backstepping control for a tank process based on adaptive observers; Aguilar-Garnica et al. [11] designed and applied a reset observer in order to provide on-line estimation of the concentration of Free Fatty Acids from temperature measurements within a continuous stirred tank reactor; Eleiwi and Laleg-Kirati [12] presented a nonlinear observer-based Lyapunov control for a membrane distillation process; Kleindienst et al. [13] used the measurement of temperature at one single point of a silicon wafer to estimate the remaining wafer temperature profile using a state observer; and Lisci et al. [14] developed a geometric observer to design a model-based soft sensor for the estimation of quality indexes in a bioreactor. Sheng et al. [15] went further and showed how smart soft sensors can be combined with machine learning techniques to significantly save human resources and improve performance under complex industrial conditions. Although there are several estimation techniques, as stated in [16], it is difficult to select the proper one for specific applications; hence, the authors performed a review of observers recently used in process control and classified them into six classes.

Several studies for control systems have been performed using tanks because of their reconfiguration possibilities and easiness to visualize variables [17,18]. Hou et al. [19] provided a method for designing nonlinear state observers that can be used to design observers for models of induction motors and three-tank systems. Hou et al. [20] performed the observability analysis and observer synthesis for a three-tank water process taking into account singularities of nonlinear observers. Pan et al. [21] addressed the nonlinear control design problem for a state-coupled two-tank liquid level system. Korbicz and Witczak [22] designed a bank of extended unknown input observers for fault diagnosis of a two-tank system. Zhou et al. [23] designed three receding horizon predictive control strategies for a three-tank system based on the structural characteristics of a hybrid pseudo-linear RBF-ARX model. Prajapati and Roy [24] used unknown input observers for fault detection and isolation in a three-coupled tank system. Li et al. [25] proposed and evaluated an energy-efficient data transmission scheme for remote state estimation in a two-tank system. More recently, Arasu and Prakash [26] designed and implemented predictor-corrector based control schemes for a single input-single output nonlinear lab-scale conical tank system. Patel and Shah [27] developed a fault-tolerant controller for nonlinear systems that was implemented and validated on a computer model of a three interconnected conical tank system. Zhao and Zhang [28] proposed an inverse tangent functional nonlinear feedback control and carried out a case study related to the water tank level control system. Most of the referenced works rely on two types of tanks: cascade tanks, where the input flow goes into the first tank, and the level of each tank depends on the level of the previous tank, such as the benchmark quadruple tank system [29,30,31], and interacting tanks, where the input flow additionally depends on the current level [32,33].

High-gain observers have been considered to estimate the state in processes with nonlinear dynamics because they have robust estimation properties and disturbance rejection capabilities [34,35,36,37,38]. Regarding process control using HGOs, some works have been reported in literature. For instance, Lafont et al. [39] designed an adaptive high-gain observer for wastewater treatment systems. Turki et al. [40] introduced an output-feedback control scheme that combines nonlinear backstepping control with an adaptive high-gain observer in a two-tank process. Banerjee and Jana [41] synthesized an estimator-based hybrid control scheme that consists of a high gain nonlinear observer and the extended generic model controller and tested it in reactive distillation column. Gouta et al. [42] reported the nonlinear control design for a state coupled two-tank liquid level system which combines a nonlinear generalized predictive controller with a high-gain observer. Ayadi et al. [43] implemented a high-gain observer which provides a full state estimation (position, velocity, temperatures and pressures) in an electropneumatic system. Wang et al. [44] proposed a sliding-mode dynamic surface control strategy based on a high-gain observer for glue mixing and dosing control in the particleboard process.

Although some reported works in the literature about observer design show plant tests in fixed lab rigs, just a few provide either the complete observability analysis or the use of robust industrial instruments. This work addresses the complete observability analysis, in the Herman–Krener sense, for a three-tank process that combines tanks in series, interaction and nonlinear weirs and the design and implementation of a high-gain observer that guarantees convergence of the state estimation. The validation and comparison to the Extended Kalman Filter, of this observer, was performed using both simulations and real tests in a multipurpose experimental station that uses industrial instrumentation and provides flexibility to change dynamics and nonlinear behavior, as briefly addressed in [18].

The organization of the paper is as follows. In Section 2, the plant is described and a state–space model is analyzed. Section 3 shows the observability analysis of the set-up using the notions in [45]. Section 4 contains the design of the high-gain observer. Numerical simulations are provided in Section 5. The experimental results for the multipurpose station are presented in Section 6. Finally, conclusions are presented in Section 7.

## 2. Process Modeling

As addressed in [18], the process comprises three tanks arranged as shown in Figure 1. The system consists of three tanks, $T_{1}$, $T_{2}$, and $T_{3}$, and each tank is equipped with a real level transmitter (LT). The two interacting Tanks $T_{1}$ and $T_{2}$ are connected in series with the third Tank $T_{3}$. An electrical pump (P-1) is used to provide water supply into the system through two different branches, namely main and secondary, and, to measure the flow through the branches, each branch is equipped with an electromagnetic flow transmitter (FIT). Flow regulation in the main branch is achieved with the help of a control valve (UV-UY), which allows the manipulation of input flow to Tank $T_{1}$. Using a manual valve (HV) located in the secondary branch, disturbances to the system are injected at Tank $T_{3}$. In this paper, the input to the system is the flow regulated by the control valve, whereas the output of the system is the level in Tank $T_{3}$ (see [46,47,48] for further details about the process).

The mathematical model can be obtained performing a mass balance in each tank, which is given by [18]
(1)$${\dot{m}}_{\mathrm{in}}\left(t\right)-{\dot{m}}_{\mathrm{out}}\left(t\right)=\dot{m}\left(t\right)$$
where ${\dot{m}}_{\mathrm{in}}$ and ${\dot{m}}_{\mathrm{out}}$ are the input and output mass flows, respectively, and $\dot{m}$ denotes the accumulation of mass in the tank. Assuming environment conditions, the density of water (the fluid process) is constant; therefore, Equation (Equation 1) becomes
(2)$$A\dot{h}\left(t\right)=q_{\mathrm{in}}\left(t\right)-q_{\mathrm{out}}\left(t\right)$$
where $q_{\mathrm{in}}$ and $q_{\mathrm{out}}$ are the input and output volumetric flows, respectively, *A* is the cross-section area and $\dot{h}$ is the level variation in that tank.

Applying Equation (Equation 2) for Tanks $T_{1}$, $T_{2}$, and $T_{3}$ yields
(3)$$\left[\begin{array}{c} A_{1}{\dot{h}}_{1}\\ A_{2}{\dot{h}}_{2}\\ A_{3}{\dot{h}}_{3} \end{array}\right]=\left[\begin{array}{c} q_{\mathrm{in}}-k_{12}\sqrt{|h_{1}-h_{2}|}\mathrm{sgn}\left(h_{1}-h_{2}\right)\\ k_{12}\sqrt{|h_{1}-h_{2}|}\mathrm{sgn}\left(h_{1}-h_{2}\right)-q_{2}\left(h_{2}\right)\\ q_{2}\left(h_{2}\right)-q_{3}\left(h_{3}\right) \end{array}\right]$$
with $q_{\mathrm{in}}$ the input flow to Tank $T_{1}$ and $q_{i}\left(h_{i}\right)$ the output flow from Tank $T_{i}$, i∈{2,3}, where $A_{j}>0$ and $h_{j}$ are the cross-section areas and water levels of Tank $T_{j}$, j∈{1,2,3}, respectively, and $k_{12}>0$ is the coupling orifice constant. In Equation (Equation 3), |·|:R→[0,∞) is the absolute value function and sgn:R→R is the sign function, that is,
$$\left|x\right|\overset{\mathrm{def}}{=}\left\{\begin{array}{cc} x & \mathrm{if}\;\;\;x\geq 0,\\ -x & \mathrm{if}\;\;\;x<0, \end{array}\right.$$
and
$$\mathrm{sgn}\left(x\right)\overset{\mathrm{def}}{=}\left\{\begin{array}{cc} -1 & \mathrm{if}\;\;\;x<0,\\ 0 & \mathrm{if}\;\;\;x=0,\\ 1 & \mathrm{if}\;\;\;x>0. \end{array}\right.$$

The output flow from Tank $T_{2}$ to Tank $T_{3}$ can be chosen according to the sharp crested weir type. The general formula for the weir [49] can be written as
(4)$$q_{j}\overset{\mathrm{def}}{=}k_{j}{\left(h_{j}-{\mathrm{lb}}_{j}\right)}^{n_{j}}H\left(h_{j}-{\mathrm{lb}}_{j}\right),\;j\in \left\{2,3\right\}$$
where $k_{j}$ is the coefficient of the weir for Tank $T_{j}$ and ${\mathrm{lb}}_{j}$ is the distance from the bottom of the tank to the crest of the weir in Tank $T_{j}$; $n_{j}$ is the order of the weir, which depends on the shape; and H(·) is the Heaviside step function, that is,
$$H\left(x\right)\overset{\mathrm{def}}{=}\left\{\begin{array}{cc} 0 & \mathrm{if}\;\;\;x\leq 0\\ 1 & \mathrm{if}\;\;\;x>0 \end{array}\right.$$

In addition, since the zeros of the sensors can be calibrated to match the lower boundary of each weir and the working space is always above this level, for each j∈{2,3}, Equation (Equation 4) can be written as $q_{j}=k_{j}h_{j}^{n_{j}},\;\;\;h_{j}>0$.

To get a state–space realization of the system, let us define the state vector as $x={\left[x_{1},x_{2},x_{3}\right]}^{T}\overset{\mathrm{def}}{=}{\left[h_{1},h_{2},h_{3}\right]}^{T},$ and define the function s↦m(s) as $m\left(s\right)\overset{\mathrm{def}}{=}\sqrt{\left|s\right|}\mathrm{sgn}\left(s\right)$. The water level in the three tanks is non-negative, i.e., $h_{i}\geq 0$, i∈{1,2,3}. Then, the state–space representation of the plant is given by
(5)$$\left.\begin{array}{cc} \dot{x} & =f\left(x\right)+g_{1}\left(x\right)u_{1}\\ y & =h(x) \end{array}\right\}$$
where the state vector $x\in R_{\geq 0}^{3}$ and the scalar input $u_{1}>0$ is the input flow, i.e., $u_{1}=q_{\mathrm{in}}$. The drift and control vector fields are given by
(6)$$\begin{array}{cc} f(x) & =AK\mu (x) \end{array}$$
(7)$$\begin{array}{cc} g_{1}\left(x\right) & =A_{1}^{-1}e_{1} \end{array}$$
with $A\overset{\mathrm{def}}{=}\mathrm{diag}\left(A_{1}^{-1},A_{2}^{-1},A_{3}^{-1}\right)$,
$$\begin{array}{cc} K & \overset{\mathrm{def}}{=}\left[\begin{array}{ccc} -k_{12} & 0 & 0\\ k_{12} & -k_{2} & 0\\ 0 & k_{2} & -k_{3} \end{array}\right] \end{array}$$
and x↦μ(x) is given by
(8)$$\begin{array}{cc} \mu (x) & \overset{\mathrm{def}}{=}\left[\begin{array}{c} m(x_{1}-x_{2})\\ x_{2}^{n_{2}}\\ x_{3}^{n_{3}} \end{array}\right] \end{array}$$
where $e_{i}\in R^{3}$, i∈{1,2,3}, is the *i*th column of the 3×3 identity matrix $I_{3}$. The output function is linear, that is, $h\left(x\right)=c^{T}x$ with $c\overset{\mathrm{def}}{=}e_{3}$. In addition, recall that $A_{1}$, $A_{2}$, $A_{3}$, $k_{12}$, $k_{2}$, $k_{3}$, $n_{2}$ and $n_{3}$ are all known, nonzero positive constants and hence *A* and *K* are nonsingular matrices. Consequently, from Equation (Equation 6), the equilibrium points are given by those points where μ(·) vanishes, that is, $X_{e}\overset{\mathrm{def}}{=}\left\{x\in R_{\geq 0}^{3}:\mu \left(x\right)=0\right\}$; from Equation (Equation 8) it follows that $X_{e}=\left\{0\right\}$.

From Figure 1, it can be noticed that the level in the third tank only depends on the output flow of the second Tank $T_{2}$. Therefore, the complete system can be seen as the interconnection of two systems in cascade: two interacting tanks and the lower tank. Hence, the phase portrait is used in order to analyze the behaviour of the two interacting Tanks $T_{1}$ and $T_{2}$. To this end, $R_{1}\overset{\mathrm{def}}{=}\left\{{\left[x_{1},x_{2},x_{3}\right]}^{T}\in R_{\geq 0}^{3}:x_{1}>x_{2}\right\}$, $R_{2}\overset{\mathrm{def}}{=}\left\{{\left[x_{1},x_{2},x_{3}\right]}^{T}\in R_{\geq 0}^{3}:x_{1}<x_{2}\right\}$, and $R_{3}\overset{\mathrm{def}}{=}\left\{{\left[x_{1},x_{2},x_{3}\right]}^{T}\in R_{\geq 0}^{3}:x_{1}=x_{2}\right\}$. Referring to Figure 2, $R_{1},R_{2}$ and $R_{3}$, respectively, denote regions where the level in $T_{1}$ is greater than that of $T_{2}$, the level in $T_{1}$ is lower than in $T_{2}$ and the levels in Tanks $T_{1}$ and $T_{2}$ are equal.

Figure 3 shows the phase portrait of the levels in Tanks $T_{1}$ and $T_{2}$ without the input. As it is shown, trajectories starting from different initial conditions in the regions $R_{1}$ and $R_{2}$ approach the equilibrium point (origin). Nevertheless, this depends on the value of $k_{12}$, since a large value correspond to almost zero restriction between the two tanks, transforming the two tanks into a single one.

## 3. Observability Analysis

Consider a general nonlinear system
(9)$$\left.\begin{array}{cc} \dot{x} & =f(x,u)\\ y & =h(x) \end{array}\right\}$$
defined on $R^{n}$, where f is a smooth and complete vector field on $R^{n}$, the input vector u takes values in a compact subset Ω of $R^{r}$ containing zero in its interior and the output function $h:R^{n}\to R^{m}$ has smooth components $h_{1},\ldots ,h_{q}$.

Two states $x_{1},x_{2}\in R^{n}$ are *indistinguishable* for the system in Equation (Equation 9) if, for every admissible input u∈Ω, the solutions of Equation (Equation 9) satisfying the initial conditions $x\left(0\right)=x_{1}$ and $x\left(0\right)=x_{2}$ produce identical output-time histories. In other words, $x_{1},x_{2}\in R^{n}$ are indistinguishable for the system in Equation (Equation 9), if and only if $h\left(\Phi_{u}\left(t,x_{1}\right)\right)=h\left(\Phi_{u}\left(t,x_{2}\right)\right)$ for every t≥0 and input u, where $\Phi_{u}\left(t,x_{0}\right)$ denotes the solution of the system in Equation (Equation 9) at time *t* for the initial condition $x_{0}$ and the input u. Given $x_{0}\in R^{n}$, let us denote $I\left(x_{0}\right)\subseteq R^{n}$ as the set of all points that are indistinguishable from $x_{0}$ with respect to the system in Equation (Equation 9).

The following definitions from [45] are used. The system in Equation (Equation 9) is *observable at*
$x_{0}\in R^{n}$ if $I\left(x_{0}\right)=\left\{x_{0}\right\}$, and it is *observable* if $I\left(x_{0}\right)=\left\{x_{0}\right\}$ for every $x_{0}$. The system in Equation (Equation 9) is *weakly observable at*
$x_{0}\in R^{n}$ if $x_{0}$ is an isolated point of $I(x_{0})$. The system in Equation (Equation 9) is *weakly observable* if it is weakly observable at every $x_{0}\in R^{n}$. Clearly, observability implies weak observability.

The following assumptions are made:

**Assumption** **1.**
*Let us assume that the plant operates in the region*
$$\chi \overset{\mathrm{def}}{=}\left\{\left(x_{1},x_{2},x_{3}\right)\in R_{\geq 0}^{3}:x_{1}>x_{2}\;\;and\;\;x_{3}>0\right\}$$


**Assumption** **2.***The input flow*$u_{1}\in R$*is bounded (above and below), i.e.,*$0\leq u_{1}\leq u_{\max}$*and enters into the plant through Tank*$T_{1}$, *where*$u_{\max}$*is the maximum input flow*.


The implications of Assumption 2 are the following. The maximum input flow $u_{\max}$ ensures no water overflow from the tanks, which is true since the plant was designed to avoid liquid overtopping the tank’s physical limit [46,47]. Additionally, water always flows from Tank $T_{1}$ to Tank $T_{2}$ and finally to Tank $T_{3}$. Then, it is assumed that there are no disturbances, i.e., the only water input into the system is located at the first tank. In addition, it ensures that the level in each tank is bounded, that is $x_{i}<L_{i_{\max}}$, i∈{1,2,3}, where $L_{i_{\max}}$ is the maximum level in the *i*th tank.

Given $x_{0}\in \chi$, let $I_{0}\left(x_{0}\right)$ denote the set of indistinguishable states from the initial condition $x_{0}$ for the unactuated system in Equation (Equation 9). Our next result characterizes this set.

**Proposition** **1.***For the unactuated system in Equation (Equation 9)*, $I_{0}\left(x_{0}\right)=\left\{x_{0}\right\}$*for every*$x_{0}\in \chi$.


**Proof.** First, observe that, in this case, u≡0. Consider $x_{0}=\left(x_{10},x_{20},x_{30}\right)\neq 0\in \chi$ and let ${\bar{x}}_{0}:=\left({\bar{x}}_{10},{\bar{x}}_{20},{\bar{x}}_{30}\right)\neq x_{0}$ be such that ${\bar{x}}_{0}\in I_{0}\left(x_{0}\right)$. Then, $h\left(\Phi_{u}\left(t,{\bar{x}}_{0}\right)\right)=h\left(\Phi_{u}\left(t,x_{0}\right)\right)$ for all $t\in [0,t_{f})$, which implies that ${\bar{x}}_{30}=x_{30}$. Now, $h\left(\Phi_{u}\left(t,{\bar{x}}_{0}\right)\right)=h\left(\Phi_{u}\left(t,x_{0}\right)\right)$ for all $t\in [0,t_{f})$ also implies that $L_{f}^{j}h\left({\bar{x}}_{0}\right)=L_{f}^{j}h\left(x_{0}\right),\;\forall \;j>0$, where $L_{f}h$ is the Lie-derivative of the function *h* along the vector field f. Recall that $L_{f}h\left(x\right)=\nabla h{\left(x\right)}^{T}f\left(x\right)$, where ∇ is the gradient operator. Using this, we have
$$L_{f}h\left(x\right)=e_{3}^{T}f\left(x\right)=e_{3}^{T}AK\mu \left(x\right)=A_{3}^{-1}\left(k_{2}x_{2}^{n_{2}}-k_{3}x_{3}^{n_{3}}\right).$$Hence, $L_{f}h\left({\bar{x}}_{0}\right)=L_{f}h\left(x_{0}\right)$ yields
$$A_{3}^{-1}\left(k_{2}{\bar{x}}_{20}^{n_{2}}-k_{3}{\bar{x}}_{30}^{n_{3}}\right)=A_{3}^{-1}\left(k_{2}x_{20}^{n_{2}}-k_{3}x_{30}^{n_{3}}\right).$$Since $A_{3}>0$, the last equation implies that $k_{2}{\bar{x}}_{20}^{n_{2}}-k_{3}{\bar{x}}_{30}^{n_{3}}=k_{2}x_{20}^{n_{2}}-k_{3}x_{30}^{n_{3}}$, and substituting ${\bar{x}}_{30}=x_{30}$ immediately implies that ${\bar{x}}_{20}=x_{20}$.Next, $L_{f}^{2}h\left({\bar{x}}_{0}\right)=L_{f}h^{2}\left(x_{0}\right)$ yields
$$\begin{array}{c} \;{\left(A_{2}A_{3}\right)}^{-1}k_{2}n_{2}{\bar{x}}_{20}^{n_{2}-1}\left(k_{12}\sqrt{{\bar{x}}_{10}-{\bar{x}}_{20}}-k_{2}{\bar{x}}_{20}^{n_{2}}\right)-A_{3}^{-2}k_{3}n_{3}{\bar{x}}_{30}^{n_{3}-1}\left(k_{2}{\bar{x}}_{20}^{n_{2}}-k_{3}{\bar{x}}_{30}^{n_{3}}\right)=\\ {\left(A_{2}A_{3}\right)}^{-1}k_{2}n_{2}x_{20}^{n_{2}-1}\left(k_{12}\sqrt{x_{10}-x_{20}}-k_{2}x_{20}^{n_{2}}\right)-A_{3}^{-2}k_{3}n_{3}x_{30}^{n_{3}-1}\left(k_{2}x_{20}^{n_{2}}-k_{3}x_{30}^{n_{3}}\right). \end{array}$$Since ${\bar{x}}_{20}=x_{20}$ and ${\bar{x}}_{30}=x_{30}$, the last equation becomes
$$\begin{array}{c} {\left(A_{2}A_{3}\right)}^{-1}k_{12}k_{2}n_{2}x_{20}^{n_{2}-1}\sqrt{{\bar{x}}_{10}-x_{20}}={\left(A_{2}A_{3}\right)}^{-1}k_{12}k_{2}n_{2}x_{20}^{n_{2}-1}\sqrt{x_{10}-x_{20}}. \end{array}$$Simplifying yields $x_{20}^{n_{2}-1}\sqrt{{\bar{x}}_{10}-x_{20}}=x_{20}^{n_{2}-1}\sqrt{x_{10}-x_{20}}$. This implies that $\sqrt{{\bar{x}}_{10}-{\bar{x}}_{20}}=\sqrt{x_{10}-x_{20}}$, and hence ${\bar{x}}_{10}=x_{10}$. Thus, it has been shown that ${\bar{x}}_{0}\in I_{0}\left(x_{0}\right)$ implies that ${\bar{x}}_{0}=x_{0}$. In other words, $I_{0}\left(x_{0}\right)\subseteq \left\{x_{0}\right\}$. This completes the proof. □

Proposition 1 implies that the unactuated system in Equation (Equation 9) is observable in the Herman–Krener sense. Given $x_{0}\in \chi$, let $I_{1}\left(x_{0}\right)$ denote the set of indistinguishable states from the initial condition $x_{0}$ for the actuated system in Equation (Equation 9). Clearly, $I_{1}\left(x_{0}\right)\subseteq I_{0}\left(x_{0}\right)=\left\{x_{0}\right\}$. Consequently, $I_{1}\left(x_{0}\right)=\left\{x_{0}\right\}$, and hence the actuated system in Equation (Equation 9) is also observable in the sense of Herman–Krener. Thus, it has been shown that the three-tank system with the knowledge of the input and the water level in Tank $T_{3}$ is observable in the sense of Herman–Krener. A high-gain observer is proposed in the following section.

## 4. Observer Design

Consider the nonlinear system of the form
(10)$$\left.\begin{array}{cc} \dot{x} & =Ax+\varphi \left(u,x\right)\\ y & =Cx \end{array}\right\}$$
where $x\in X\subset R^{n}$ denotes the state vector, $u:\left[0,T\right]\to U\subset R^{r}$ is the input vector, $y\in Y\subset R^{m}$ is the output vector,
$$\begin{array}{cc} A & \overset{\mathrm{def}}{=}\left[\begin{array}{cccc} 0 & 1 & & 0\\ \vdots & & \ddots & \\ & & & 1\\ 0 & & \cdots & 0 \end{array}\right]\in R^{n\times n},\\ C & \overset{\mathrm{def}}{=}\left[\begin{array}{cccc} 1 & 0 & \cdots & 0 \end{array}\right]\in R^{n}, \end{array}$$
and $\varphi :X\times U\to R^{n}$ is of the form
$$\varphi \left(u,x\right)\overset{\mathrm{def}}{=}\left[\begin{array}{c} \varphi_{1}\left(u,x_{1}\right)\\ \vdots \\ \varphi_{n}\left(u,x_{1},\cdots ,x_{n}\right) \end{array}\right],$$
which is a globally Lipschitz nonlinear function that contains the nonlinearities of the system.

Consider θ>0 and let $k\in R^{n}$ be such that the matrix A−kC is Hurwitz. Referring to Besançon [50], for the system in Equation (Equation 10), the observer is given by
(11)$$\dot{\hat{x}}=A\hat{x}+\varphi \left(u,\hat{x}\right)-\Delta_{\theta }k\left(C\hat{x}-y\right),$$
where $\Delta_{\theta }\overset{\mathrm{def}}{=}\mathrm{diag}\left(\theta ,\ldots ,\theta^{n}\right)\in R^{n\times n}$. This form of observer is called *high-gain* observer (HGO), and it has good robust estimation properties and disturbance rejection capabilities [51]. The idea is to use a high-gain constant in order to generate a fast response in the observer and reduce the effect of the nonlinear error. Nevertheless, due to the high gain, the observer is very sensitive to noise and the transient response, when the estimation is far from the true value, can cause peaks in the prediction. However, this drawback can be solved by filtering the output signal or introducing a saturation function in the feedback control law [52,53], making them a good solution for state estimation.

To obtain an observer of the form of Equation (Equation 11), the system in Equation (Equation 5) must be transformed into its observability canonical form. According to Besançon [54], any control affine system satisfying the observability rank condition can be turned into the form Equation (Equation 10) using a diffeomorphism $z:R^{n}\to R^{n}$ given by
$$z\left(x\right):=\left[\begin{array}{c} z_{1}\left(x\right)\\ \vdots \\ z_{n}\left(x\right) \end{array}\right],$$
where $z_{k}\left(x\right)=L_{f}^{k-1}h\left(x\right),\;\;k=1,\ldots ,n$, with $L_{f}^{0}h\overset{\mathrm{def}}{=}h$.

Thus, the control affine system is related to its observability canonical form. To design an observer for the original plant and to avoid the inverse transformation of the observer obtained for the canonical form, it is recalled the definition of system equivalence used in [50].

**Definition** **1.***(System equivalence).**For each*i∈{1,2}, *consider the systems*$$\sum_{i}:\left\{\begin{array}{c} \begin{array}{ccc} {\dot{x}}_{i} & = & f_{i}\left(x_{i},u_{i}\right),\\ y_{i} & = & h_{i}\left(x_{i}\right), \end{array} \end{array}\right.$$*defined on*$R^{n_{i}}$*with input*$u_{i}\in R^{m_{i}}$, *output*$y_{i}\in R^{p}$*and*$f_{i}$’s *are smooth vector fields*. *The two systems*$\sum_{1}$*and*$\sum_{2}$*are (state–space)* equivalent *if there exists a diffeomorphism*
$z:R^{n_{1}}\to R^{n_{2}}$
*such that*
$\left(x_{2},u_{2}\right)=\left(z\left(x_{1}\right),\left(u_{1}\right)\right)$.

In other words, if $\left(x_{1}\left(t\right),u\left(t\right)\right)$ is a trajectory of $\sum_{1}$, then $\left(z\left(x_{1}\left(t\right)\right),u\left(t\right)\right)$ is a trajectory of $\sum_{2}$, that is,
$${\dot{x}}_{2}=\frac{d}{dt}z\left(x_{1}\right)=\frac{\partial z}{\partial x_{1}}f_{1}\left(x_{1},u\right)=f_{2}(z\left(x_{1}\right),u).$$

Consider the systems $\sum_{1}$ and $\sum_{2}$ and suppose
$$O_{1}:\left\{\begin{array}{c} \begin{array}{ccc} {\dot{\hat{x}}}_{2} & = & f_{2}\left({\hat{x}}_{2},u\right)+k\left(w,h_{2}\left({\hat{x}}_{2}\right)-y_{2}\right)\\ \dot{w} & = & f_{w}\left(w,u,y_{2}\right) \end{array} \end{array}\right.$$
is an observer for the system $\Sigma_{2}$. Then,
$$O_{2}:\left\{\begin{array}{c} \begin{array}{ccc} {\dot{\hat{x}}}_{1} & = & f_{1}\left({\hat{x}}_{1},u\right)+{\left({\left.\frac{\partial \Phi }{\partial x}\right|}_{\hat{x}}\right)}^{-1}k\left(w,h_{1}\left({\hat{x}}_{1}\right)-y_{1}\right)\\ \dot{w} & = & f_{w}\left(w,u,y_{1}\right) \end{array} \end{array}\right.$$
is an observer for the system $\sum_{1}$. Using the concept of system equivalence and by transforming the system into its observability canonical form, a HGO is proposed for the three-tank system.

**Proposition** **2.**
*Consider the three-tank system in Equation (Equation 5) and suppose that Assumption 1 holds. Define*
$\Delta_{\theta }=\mathrm{diag}\left(\theta ,\theta^{2},\theta^{3}\right)$
*with*
$\theta \in R_{>0}$
*and let*
$k={\left[k_{1},k_{2},k_{3}\right]}^{T}\in R^{3}$
*be such that*
$A-kc^{T}$
*is Hurwitz. Then*
(12)$$\begin{array}{cc} \dot{\hat{x}} & =f\left(\hat{x}\right)+g_{1}\left(\hat{x}\right)u_{1}+{\left.{\left(\frac{\partial \Phi (x)}{\partial x}\right)}^{-1}\right|}_{\hat{x}}\Delta_{\theta }k\left(c^{T}\hat{x}-y\right), \end{array}$$

*is an asymptotic observer for the system in Equation (Equation 5), where*
$$\begin{array}{cc} \Phi \left(x\right)= & \left[\begin{array}{c} x_{3}\\ A_{3}^{-1}\left(k_{2}x_{2}^{n_{2}}-k_{3}x_{3}^{n_{3}}\right)\\ {\left(A_{2}A_{3}\right)}^{-1}k_{2}n_{2}x_{2}^{n_{2}-1}\left(k_{12}\sqrt{x_{1}-x_{2}}-k_{2}x_{2}^{n_{2}}\right)-A_{3}^{-2}k_{3}n_{3}x_{3}^{n_{3}-1}\left(k_{2}x_{2}^{n_{2}}-k_{3}x_{3}^{n_{3}}\right) \end{array}\right]. \end{array}$$


**Proof.** To check the observability rank condition, the Jacobian matrix of Φ(x) must be full rank. The Jacobian is given by
$$J_{\Phi }\left(x\right)=\left[\begin{array}{ccc} 0 & 0 & 1\\ 0 & \alpha_{1} & \alpha_{2}\\ \alpha_{3} & \alpha_{4} & \alpha_{5} \end{array}\right],$$
where
$$\begin{array}{cc} \alpha_{i}= & {\left(-1\right)}^{i+1}A_{3}^{-1}k_{i+1}n_{i+1}x_{i+1}^{n_{i+1}-1},\;i\in \left\{1,2\right\},\\ \alpha_{3}= & \frac{1}{2}{\left(A_{2}A_{3}\right)}^{-1}k_{2}n_{2}x_{2}^{n_{2}-1}k_{12}{\left(x_{1}-x_{2}\right)}^{-\frac{1}{2}}\\ \alpha_{4}= & {\left(A_{2}A_{3}\right)}^{-1}k_{2}n_{2}\left(n_{2}-1\right)x_{2}^{n_{2}-2}\left(k_{12}\sqrt{x_{1}-x_{2}}-k_{2}x_{2}^{n_{2}}\right)\\ & +{\left(A_{2}A_{3}\right)}^{-1}k_{2}n_{2}x_{2}^{n_{2}-2}\left(-\frac{1}{2}k_{12}{\left(x_{1}-x_{2}\right)}^{-\frac{1}{2}}\right.\\ & \left.-k_{2}n_{2}x_{2}^{n_{2}-1}\right)-A_{3}^{-2}k_{3}n_{3}x_{3}^{n_{3}-1}\left(k_{2}n_{2}x_{2}^{n_{1}-2}\right)\\ \alpha_{5}= & -A_{3}^{-2}k_{3}n_{3}\left(n_{3}-1\right)x_{3}^{n_{3}-2}\left(k_{2}x_{2}^{n_{2}}-k_{3}x_{3}^{n_{3}}+A_{3}^{-2}k_{3}^{2}n_{3}^{2}x_{3}^{2(n_{3}-1)}\right) \end{array}$$Thus, $J_{\Phi }\left(x\right)$ is nonsingular if and only if $\alpha_{1}\alpha_{3}\neq 0$, which is true if and only if $0<x_{2}<x_{1}$. Under Assumption 1, it now follows that the Jacobian has full rank.Applying the diffeomorphism z=Φ(x) to the observer in Equation (Equation 12) yields
(13)$$\begin{array}{c} \dot{\hat{z}}={\left.\frac{\partial \Phi (x)}{\partial x}\right|}_{\hat{x}}\left[f\left(\hat{x}\right)+g_{1}\left(\hat{x}\right)u_{1}\right]+\Delta_{\theta }k\left(c_{0}^{T}\hat{z}-y_{2}\right), \end{array}$$
where $c_{0}=e_{1}$ and $y_{2}=c_{0}^{T}z$. According to Besançon [54],
(14)$$\begin{array}{c} {\left.\frac{\partial \Phi (x)}{\partial x}\right|}_{\hat{x}}\left[f\left(\hat{x}\right)+g_{1}\left(\hat{x}\right)u_{1}\right]=A_{0}\hat{z}+\varphi \left(\hat{z},u_{1}\right), \end{array}$$
where $\varphi \left(\hat{z},u_{1}\right)={\left[0,0,\xi \left(\hat{z},u_{1}\right)\right]}^{T}$ is Lipschitz with respect to $\hat{z}$, that is, there exists a $L_{\xi }>0$ such that $|\xi \left({\hat{z}}_{1},u_{1}\right)-\xi \left({\hat{z}}_{2},u_{1}\right)|\leq L_{\xi }∥{\hat{z}}_{1}-{\hat{z}}_{2}∥$ for all ${\hat{z}}_{1},{\hat{z}}_{2}$, and
$$A_{0}=\left[\begin{array}{ccc} 0 & 1 & 0\\ 0 & 0 & 1\\ 0 & 0 & 0 \end{array}\right].$$Using Equation (Equation 14) in Equation (Equation 13) yields
$$\dot{\hat{z}}=A_{0}\hat{z}+\varphi \left(\hat{z},u_{1}\right)+\Delta_{\theta }k(c_{0}^{T}\hat{z}-y_{2}).$$Next, define $\epsilon \overset{\mathrm{def}}{=}\Delta_{\theta }^{-1}\left(\hat{z}-z\right)$. Then, the error dynamics is given by
(15)$$\begin{array}{c} \dot{\epsilon }=\theta \left(A_{0}+kc_{0}^{T}\right)\epsilon +\Delta_{\theta }^{-1}\tilde{\varphi }\left(z,\hat{z},u_{1}\right), \end{array}$$
where $\tilde{\varphi }\left(z,\hat{z},u_{1}\right)\overset{\mathrm{def}}{=}\varphi \left(\hat{z},u_{1}\right)-\varphi \left(z,u_{1}\right)$. Inspired by the work of Hann et al. [55], the following Lyapunov function is defined
$$\begin{array}{c} V\overset{\mathrm{def}}{=}\epsilon^{T}P\epsilon , \end{array}$$
where $P=P^{T}≻0$ is such that
(16)$$\begin{array}{c} {\left(A_{0}+kc_{0}^{T}\right)}^{T}P+P\left(A_{0}+kc_{0}^{T}\right)=-I_{n}, \end{array}$$
with $I_{n}$ being the identity matrix of size n. This quadratic form was chosen in order to be positive definite and radially unbounded. Then,
(17)$$\begin{array}{c} \dot{V}={\dot{\epsilon }}^{T}P\epsilon +\epsilon^{T}P\dot{\epsilon }. \end{array}$$Using Equation (Equation 15) in Equation (Equation 17) yields
$$\begin{array}{c} \dot{V}=\theta \epsilon^{T}\left[{\left(A_{0}+kc_{0}^{T}\right)}^{T}P+P\left(A_{0}+kc_{0}^{T}\right)\right]\epsilon +2\epsilon^{T}P\Delta_{\theta }^{-1}\tilde{\varphi }\left(z,\hat{z},u_{1}\right), \end{array}$$
while substituting Equation (Equation 16) in the above yields
$$\begin{array}{c} \dot{V}=-\theta {∥\epsilon ∥}^{2}+2\epsilon^{T}P\Delta_{\theta }^{-1}\tilde{\varphi }\left(z,\hat{z},u_{1}\right), \end{array}$$
where ∥·∥ is the Euclidean norm in $R^{n}$. Since $\varphi (z,u_{1})$ is Lipschitz with respect to z, it follows that
$$∥\Delta_{\theta }^{-1}\tilde{\varphi }\left({\hat{z}}_{1},{\hat{z}}_{2},u_{1}\right)∥=\theta^{-3}|\xi \left({\hat{z}}_{1},u_{1}\right)-\xi \left({\hat{z}}_{2},u_{1}\right)|\leq \theta^{-3}L_{\xi }∥{\hat{z}}_{1}-{\hat{z}}_{2}∥\leq L_{\xi }∥\Delta_{\theta }^{-1}∥\;∥{\hat{z}}_{1}-{\hat{z}}_{2}∥$$
and hence
$$∥P\Delta_{\theta }^{-1}\tilde{\varphi }\left({\hat{z}}_{1},{\hat{z}}_{2},u_{1}\right)∥\leq L_{\xi }∥P∥\;∥\epsilon ∥.$$Thus,
$$\begin{array}{cc} \dot{V} & \leq -{\theta ∥\epsilon ∥}^{2}+2L_{\xi }{∥P∥\;∥\epsilon ∥}^{2}=-(\theta -2L_{\xi }{∥P∥)∥\epsilon ∥}^{2}. \end{array}$$By selecting $\theta >2L_{\xi }∥P∥$, it follows that the $\dot{V}<0$. This completes the proof. □

## 5. Simulation Results

The first validation stage of the HGO was performed with simulations for full-state estimation in system. Such simulations were carried out using MATLAB^®^, which is a high-performance language developed by MathWorks^®^. To get similar behavior to the real plant, a sample period of 0.2
s was selected. Different nonlinearities were tested by selecting different kind of weirs that are available for the configuration of the plant: linear, v-notch, rectangular and circular (see Figure 4).

Constant parameters were obtained measuring tanks’ areas and performing an open-loop test to get the coefficient and the exponential of Equation (Equation 4) for each weir (Table 1 and Table 2). The input flow was set to a constant value of ${\bar{q}}_{\mathrm{in}}$, and levels were measured for each steady-state condition. Then, the weirs’ parameters were computed using the MATLAB^®^ fitting toolbox, Figure 5. For the simulation, sensor noise was implemented through a Gaussian distribution with 0.5 cm standard deviation. Additionally, a deviation of 5% in all the parameters of the system was taken into account.

Step inputs of 25% of the total flow capacity were simulated every 50s approximately. The gain $k={\left[0.8,0.17,0.01\right]}^{T}$ was chosen such that $A_{0}-kc_{0}^{T}$ is Hurwitz and θ=2 as the high-gain observer parameter. Two different sets of weirs were used at the outputs of Tanks $T_{2}$ and $T_{3}$. The first set comprised a rectangular weir in the upper tank and a linear in the lower, and the second set comprised a v-notch weir in the upper tank and a circular in the lower.

Figure 6 shows the estimation of the level in each tank for the first set of weirs. Figure 7 shows the estimation of the level in each tank for the second set of weirs. The same parameters were used in both simulations, but the observer’s parameters changed according to the exponential and coefficient of each weir. The observer converges slowly to the desired value, since poles were placed near to the imaginary axis. In both simulations, level estimations are affected by noise, as expected for a HGO.

## 6. Experimental Results

The second stage of validation of the HGO was performed through full-state estimation in the real process. The experimental set-up is shown in Figure 8. This three-tank system has different control technologies: PLC, industrial controller and PC with LabVIEW^®^. In this case, the PLC controller was used as the DAQ system, and is connected through Ethernet to a LabVIEW^®^ interface [48], in which the compensator (controller+observer) was implemented.

Since the real plant is a complete experimental station, all variables were measured with real sensors, as indicated in Figure 1; hence, the state estimation can be compared to real measurements of the levels in all the tanks. The sample time for the PLC and LabVIEW^®^ program was set to 0.2
s. As in simulations, step inputs of 25% of the total flow capacity were manually performed every 50 s approximately. The gain $k\in R^{3}$ was again chosen such that $A_{0}-kc_{0}^{T}$ is Hurwitz and θ=2 as the high-gain observer parameter. Two different sets of weirs were used at the outputs of Tanks $T_{2}$ and $T_{3}$. The first set comprised a rectangular weir in the upper tank and a linear in the lower, and the second set comprised a v-notch weir in the upper tank and a circular in the lower one.

### 6.1. High-Gain Observer Performance

Figure 9 and Figure 10 show the estimation of the level in each tank for the first and second set of weirs, respectively. As can be noticed, the observer converges to the measurements. In both experiments, level estimations are affected by noise, but the level estimation for Tank $T_{3}$, considered as the output (controlled variable) behaves very similar to the real measurement. Noise in the real process is mainly caused by the configuration of the system, in which several waves are produced by the input flow (see Figure 8). Two more experiments were conducted using an HGO with higher gain, $k={\left[1.2,0.5,0.1\right]}^{T}$. Figure 11 and Figure 12 show how the state estimation is affected in a more noticeable way.

After t=200s, the system enters into an unobservable zone of the state space, and the estimation provided by the HGO diverges. This happens because the level in Tank $T_{2}$ goes below the minimum height of the weir, and, therefore, there is not flow from Tank $T_{2}$ to Tank $T_{3}$. A faster observer response can be achieved by increasing the gain k, which can result in a noisy estimation of the state, but it can still perform well when used in a complete compensator that includes the state feedback.

### 6.2. Robustness Analysis

To test the robustness of the observer, two different experiments were carried out. In the first experiment, a variation of 30% of all parameters was made (tanks areas and weir constants). In this scenario, the HGO was compared to the well-known Luenberger observer [56] and the Extended Kalman Filter (EKF) [57].

Table 3 shows the mean squared error for the three observers; as can be noticed, the HGO obtained the best performance in the real process for the complete range of the tank levels despite the 30% variation in the model parameters. Figure 13, Figure 14 and Figure 15 show the estimation for all levels with each observer. The HGO shows better performance and robustness than the other two observers. For instance, it can be seen how, for the low-level section, the estimation provided by the HGO converges in less than 2 s while Luenberger and EKF estimations are deviated from the measurement and converge after more than 20 s. It is important to point out that asymptotic estimation has been proved for the HGO, which is not the case for the EKF.

In the second experiment, an unmeasured temporary input flow went into the first and second tanks. Figure 16 and Figure 17 show the estimation of the level in each tank for a temporary input flow to the first and second tank, respectively. As was expected, the output of the system changed (level in Tank 3), and therefore the estimation should change. Recall that the level of tank *i* depends on the inflow of the previous tank, which depends on the level of such tank. Therefore, it can been seen in Figure 16 that the estimated level in Tank 2 is similar to the actual value, but higher for Tank 1. Now, when the perturbation was introduced in Tank 1, as was expected, the estimation converges to the real values of all levels (see Figure 17).

## 7. Conclusions

This paper addresses the design and implementation of a high-gain observer (HGO) for a nonlinear interacting level control process with a level measurement in one of the tanks. For this set-up, the observability analysis in the sense of Herman–Krener is presented. The observation space depends on the minimum level of each weir and the flow from Tank $T_{2}$ to Tank $T_{3}$. The notion of system equivalence is used to design an observer by taking into account the nonlinear dynamics of the system. These nonlinear effects are mainly due to the weirs, the interaction between the first two tanks and the minimum level of each weir, which defines the observable part of the state space. Theoretical and experimental results show that the HGO can provide a good estimate of the level in each tank using only the measurement of the output, when compared to other well-known linear (Luenberger) and nonlinear (EKF) state estimators; recall that asymptotic estimation has been proved for the HGO, which is not the case for the EKF. However, such estimation can be affected by noise in the measurement, which can be physically filtered using attenuators for the input flow. A faster observer dynamics specification can increase the variability of the estimation.

## Figures and Tables

**Figure 1 sensors-20-06738-f001:**
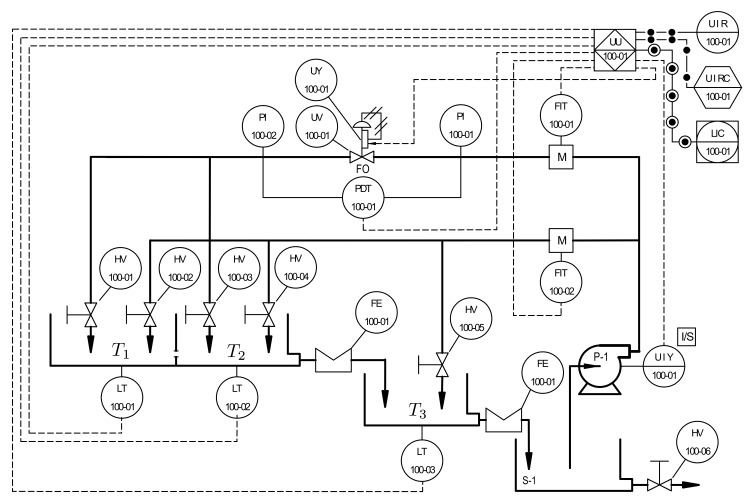
Simplified process and instrumentation diagram (P&ID) [18,47].

**Figure 2 sensors-20-06738-f002:**
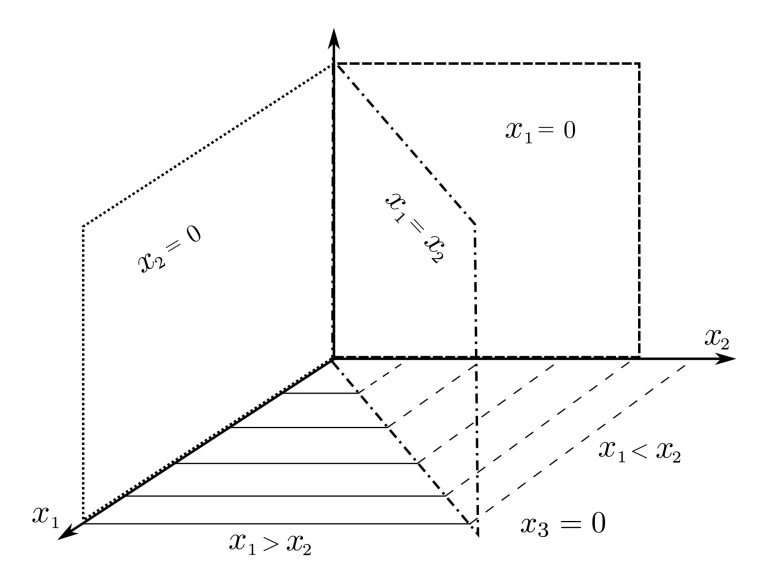
Geometrical representation $R_{1},$$R_{2}$ and $R_{3}$.

**Figure 3 sensors-20-06738-f003:**
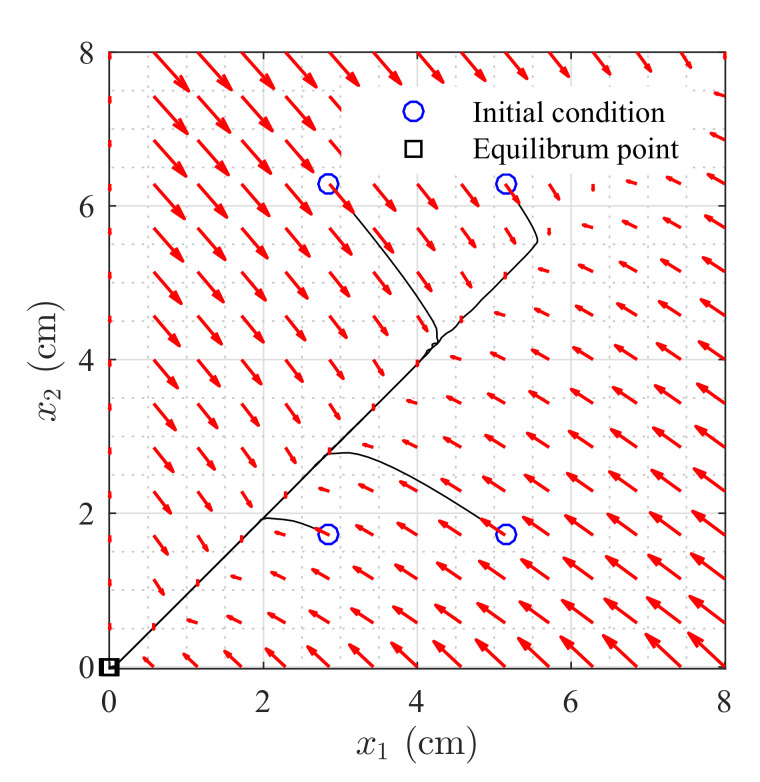
Phase portrait of the first two components of the state.

**Figure 4 sensors-20-06738-f004:**
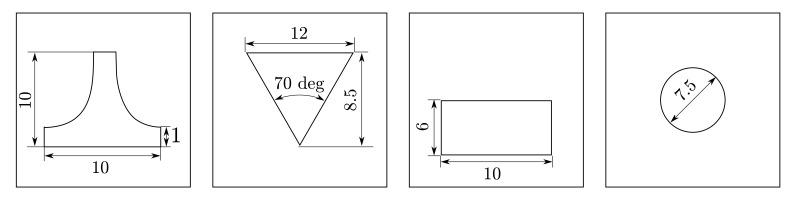
Sharp crested weirs (dimensions in cm): linear, v-notch, rectangular and circular.

**Figure 5 sensors-20-06738-f005:**
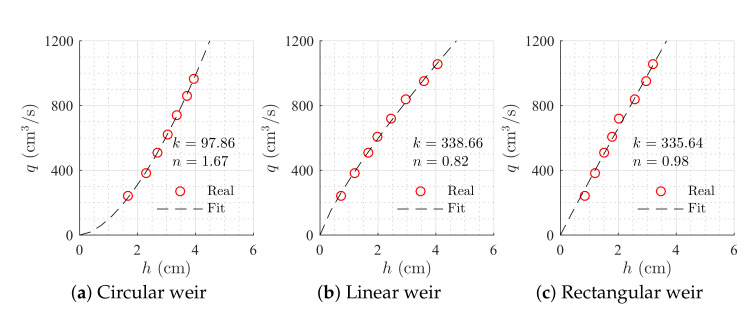
Weirs’ parameters approximation.

**Figure 6 sensors-20-06738-f006:**
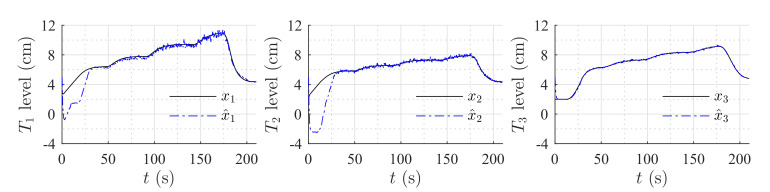
Simulation with rectangular weir at $T_{2}$ and linear weir at $T_{3}$.

**Figure 7 sensors-20-06738-f007:**
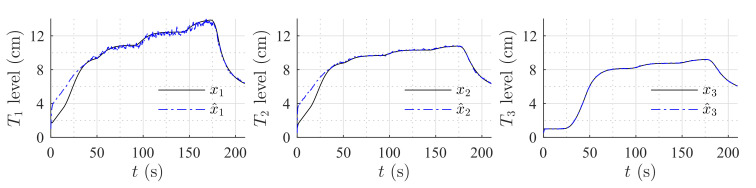
Simulation with v-notch weir at $T_{2}$ and circular weir at $T_{3}$.

**Figure 8 sensors-20-06738-f008:**
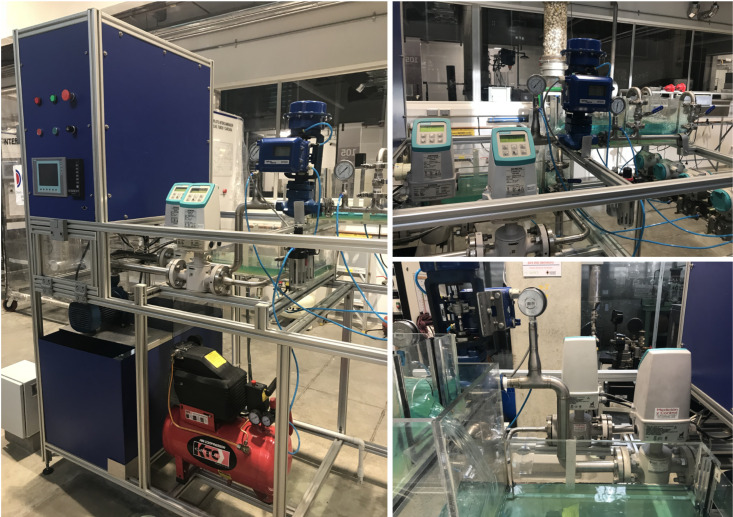
Multipurpose experimental station.

**Figure 9 sensors-20-06738-f009:**
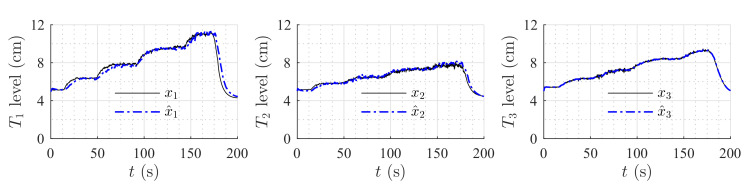
HGO experimental results: rectangular weir at $T_{2}$ and linear weir at $T_{3}$, $k={\left[0.8,0.17,0.01\right]}^{T}$.

**Figure 10 sensors-20-06738-f010:**
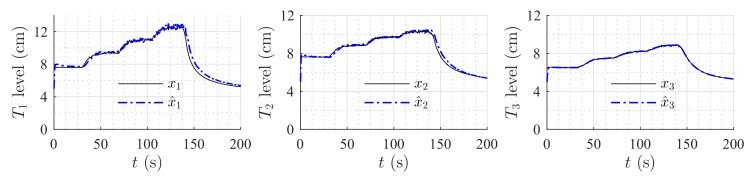
HGO experimental results: v-notch weir at $T_{2}$ and circular weir at $T_{3}$, $k={\left[0.8,0.17,0.01\right]}^{T}$.

**Figure 11 sensors-20-06738-f011:**
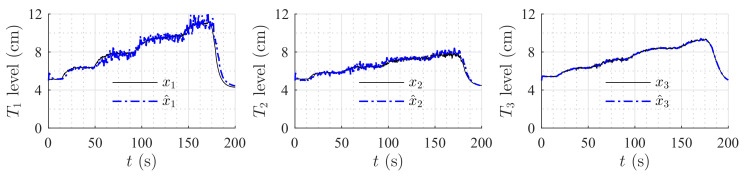
HGO experimental results: rectangular weir at $T_{2}$ and linear weir at $T_{3}$, $k={\left[1.2,0.5,0.1\right]}^{T}$.

**Figure 12 sensors-20-06738-f012:**
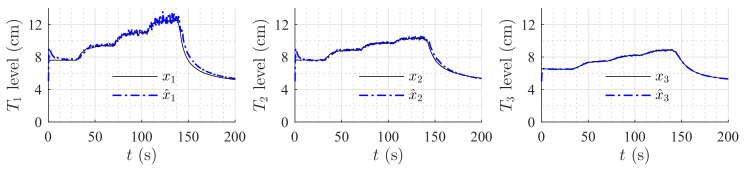
HGO experimental results: v-notch weir at $T_{2}$ and circular weir at $T_{3}$, $k={\left[1.2,0.5,0.1\right]}^{T}$.

**Figure 13 sensors-20-06738-f013:**
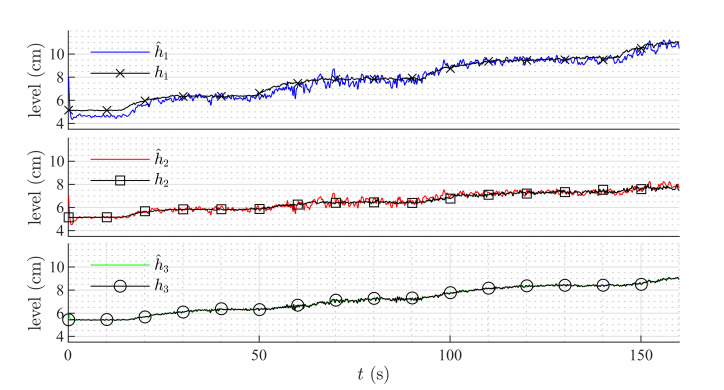
Luenberger observer performance: 30% variation in all constant parameters.

**Figure 14 sensors-20-06738-f014:**
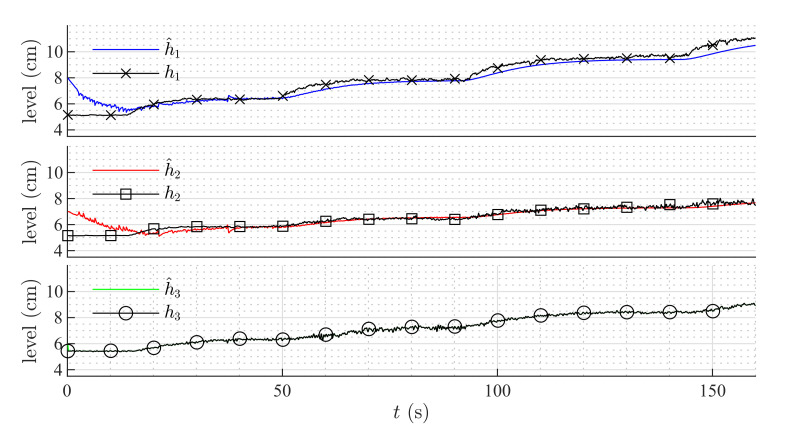
Extended Kalman Filter performance: 30% variation in all constant parameters.

**Figure 15 sensors-20-06738-f015:**
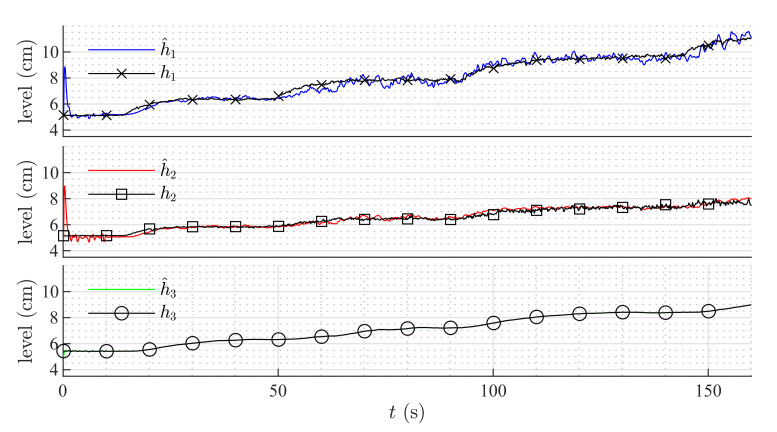
High-Gain Observer performance: 30% variation in all constant parameters.

**Figure 16 sensors-20-06738-f016:**
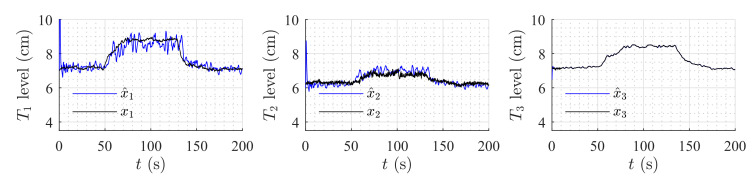
HGO estimation with unmeasured perturbation in Tank 1.

**Figure 17 sensors-20-06738-f017:**
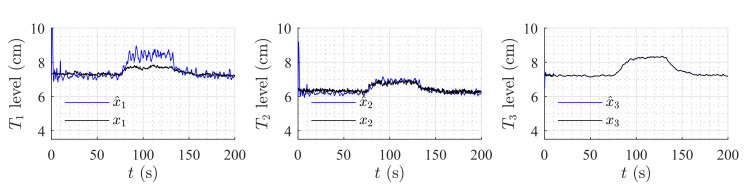
HGO estimation with unmeasured perturbation in Tank 2.

**Table 1 sensors-20-06738-t001:** Tanks’ constant parameters.

Tank 1			Tank 2			Tank 3		
$A_{1}$	764.40	cm^2^	$A_{2}$	817.60	cm^2^	$A_{3}$	1548.4	cm^2^

**Table 2 sensors-20-06738-t002:** Weirs’ constant parameters.

Parameter	Linear	V-Notch	Rectangular	Circular
*n*	0.82	2.59	0.98	1.67
*k*	338.66	7.99	335.64	97.86
lb (cm)	4.74	3.93	4.81	1.5

**Table 3 sensors-20-06738-t003:** Mean squared error (MSE) comparison.

	MSE $h_{1}$	MSE $h_{2}$	MSE $h_{3}$
	(cm${}^{2}$)	(cm${}^{2}$)	(cm${}^{2}$)
Luenberger	0.2889	0.1614	0.0371
EKF	0.3552	0.1999	0.0070
HGO	0.2243	0.1437	0.0024

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
