# Peer review of "Observability Analysis and Observer Design for a Nonlinear Three-Tank System: Theory and Experiments"

_sensors, 2020, doi:10.3390/s20236738_

Round 1

Reviewer 1 Report

the paper’s quality is generally good. Theoretical derivation is correct, both simulations and experimental validations were provided. Comments are

  1. The smooth condition for the system dynamics f and h seems too strong. Is continuous a better assumption?
  2. In the theoretical manner, the uncertainty bounds of system models are not clarified. How much robustness margin can be achieved on this manner?
  3. The estimation comparisons among couple methods are presented. How to justify the comparison is fair?  Gains are the same? Which one is more robust?
  4. In the experimental comparison, it is better to use a table to list some quantities to clarify the conclusion, like the root mean square error, convergence time, etc. Fig. 13 needs to be zoomed out.

Author Response

Please find the response within the attached document.

Reviewer 2 Report

Please see the attached pdf file.

Author Response

(The authors gave the same response as above.)

Round 2

Reviewer 1 Report

the authors have revised to address the comments. Quality has been improved. It is eligible for acceptance now.